# Cortical oscillatory dysfunction in Parkinson disease during movement activation and inhibition

**Elizabeth A. Disbrow**[1,2]\*, **Nathaniel D. Glassy**[1], **Elizabeth M. Dressler**[1],
**Kimberley Russo**[3], **Elizabeth A. Franz**[4], **Robert S. Turner**[5], **Maria I. Ventura**[6],
**Leighton Hinkley**[7], **Richard Zweig**[1,2], **Srikantan S. Nagarajan**[7], **Christina R. Ledbetter**[1,8],
**Karen A. Sigvardt**[9†]

1 LSU Health Shreveport Center for Brain Health, Shreveport, Louisiana, United States of America,
2 Department of Neurology, LSU Health Shreveport, Shreveport, Louisiana, United States of America,
3 Department of Psychology, UC Berkeley, Berkeley, California, United States of America, 4 Action Brain
and Cognition Laboratory, Department of Psychology, and fMRIotago, University of Otago, Dunedin, New
Zealand, 5 Department of Neurobiology and Center for the Neural Basis of Cognition University of Pittsburgh,
Pittsburgh, Pennsylvania, United States of America, 6 Department of Psychiatry, UC Davis, Sacramento,
California, United States of America, 7 Department of Radiology and Biomedical Imaging, University of
California, San Francisco, California, United States of America, 8 Department of Neurosurgery, LSU Health
Shreveport, Shreveport, Louisiana, United States of America, 9 Department of Neurology, UC Davis,
Sacramento, California, United States of America

† Deceased.
\* elizabeth.disbrow@lsuhs.edu

pone.0257711

University Julio de Mesquita Filho: Universidade
Estadual Paulista Julio de Mesquita Filho, BRAZIL

**Data Availability Statement:** Data for this study
has been made publicly available under Creative
Commons CC0 1.0 Universal (CC0 1.0) at

## Abstract

Response activation and inhibition are functions fundamental to executive control that are dis-
rupted in Parkinson disease (PD). We used magnetoencephalography to examine event
related changes in oscillatory power amplitude, peak latency and frequency in cortical net-
works subserving these functions and identified abnormalities associated with PD. Partici-
pants (N = 18 PD, 18 control) performed a cue/target task that required initiation of an un-
cued movement (activation) or inhibition of a cued movement. Reaction times were variable
but similar across groups. Task related responses in gamma, alpha, and beta power were
found across cortical networks including motor cortex, supplementary and pre- supplemen-
tary motor cortex, posterior parietal cortex, prefrontal cortex and anterior cingulate. PD-related
changes in power and latency were noted most frequently in the beta band, however, abnor-
mal power and delayed peak latency in the alpha band in the pre-supplementary motor area
was suggestive of a compensatory mechanism. PD peak power was delayed in pre-supple-
mentary motor area, motor cortex, and medial frontal gyrus only for activation, which is consis-
tent with deficits in un-cued (as opposed to cued) movement initiation characteristic of PD.

## Introduction

Response initiation and inhibition are functions fundamental to executive control that are dis-
rupted in Parkinson disease (PD) [1–3]. Response initiation, sometimes referred to as

https://doi.org/10.12751/g-node.nb35ss. The data
is stored in BIDS format using the CTF * .ds filetype.
It can be opened in either proprietary CTF software
(Omega 2000) or open-source M/EEG software
packages (NUTMEG, fieldtrip). The Fieldtrip
website expands on the file format here: https://
www.fieldtriptoolbox.org/getting_started/ctf/.

**Funding:** This study was funded by a grant
(RO1NS064040) from the National Institute of
Neurological Disorders and Stroke awarded to
EAD, and provided partial salary support to EAD
and LH. The funders had no role in study design,
data collection and analysis, decision to publish, or
preparation of the manuscript.

**Competing interests:** The authors have declared
that no competing interests exist.

'movement activation,' refers to the process of eliciting a desired response, whereas response
inhibition refers to the capacity to suppress inappropriate or irrelevant responses that are pre-
potent. People with PD often show deficits on standard neuropsychological tests of inhibition,
such as the Stroop Test [4, 5], which is consistent with the idea that inhibition plays a role in
automatic behaviors and action override, functions that are also disrupted in PD [6]. Further-
more, impairments in response activation and inhibition have been proposed to subserve
some of the common motor signs of PD. For example, akinesia, or an inability to start move-
ment [7–10] is consistent with a problem in movement activation [3]. Activation and inhibi-
tion are also associated with motor switching and sequencing, which are impaired in PD [3,
11]. However, deficits in response activation and inhibition associated with PD are complex
and not fully understood. For instance, in PD, the latency to initiate movement can be short-
ened by the use of pre-movement external auditory or visual cueing such as walking in time to
a metronome or walking paced by floor markers (e.g., [12–16]).

Many behavioral studies have shown that people with PD display deficits in both the activa-
tion and inhibition of motor responses [17–19]. Studies targeting components of responding
traditionally use two types of tasks, the classic 'Go Nogo' tasks [20] and 'stop-signal' tasks [21].
Wu and colleagues [22] found that people with PD performing a Go Nogo task demonstrated
increased reaction times for 'Go' trials and increased errors for both 'Go' and 'Nogo' trials as
compared to controls. Franz and Miller [23] found that people with mild to moderate PD
demonstrated abnormal force output during inhibition of 'Nogo' responses in comparison to
controls, despite a lack of difference in mean reaction times between PD and control groups.
Previously, we [1] also observed an absence of statistically significant reaction time differences
between PD and control groups during movement activation and inhibition in PD (though
variability was high) despite deterioration in spatial and temporal specificity of blood-oxygen-
level-dependent imaging signal in cortical and subcortical regions of interest.

Anatomical evidence indicates that motor and cognitive function are subserved by anatom-
ically segregated parallel circuits from the cerebral cortex to thalamus via the basal ganglia
nuclei which form circuits with projections back to the cortex. Segregated parallel loops are
thought to mediate distinct motor, cognitive, and limbic functions based on cortical targets
[24–26]. For example, abnormalities in motor cortical activity observed in PD [27, 28] have
been linked to loss of dopamine in the posterior putamen [29, 30]. Similarly, studies of execu-
tive dysfunction in PD suggest that loss of dopamine in the caudate results in abnormalities in
activity in prefrontal cortex (PFC; [31–33] for review). This PFC-connected cognitive circuit
has also been implicated in important aspects of both response activation and inhibition
behaviors ([34] for review).

There is evidence in the electroencephalography (EEG) literature of delayed onset and
decreased power in the event related responses for components of movement activation and
inhibition in people with PD [35–39], though not all studies agree (see [40] for review). Fur-
thermore, growing evidence suggests that parkinsonian impairments in motor and cognitive
function are associated with dysregulated cortical oscillatory activity. Oscillatory activity is
thought to reflect modulation of neuronal excitability in synchronously active neural assem-
blies [41, 42]. For example, using magnetoencephalography (MEG), a diffuse slowing of oscil-
latory activity at rest has been observed in people with PD, as well as increased beta band
power over sensorimotor cortex at rest [43]. Such changes are reported early in disease pro-
gression, and can be measured regardless of disease duration, stage, severity, or medication
level [44]. A number of studies demonstrate that in PD, global changes in oscillatory power are
correlated with cognitive dysfunction as well (for a review, see [45–49]). Attenuated desyn-
chronization of both beta band [22] and higher alpha band [50] EEG has been reported in peo-
ple with PD during Go and Nogo trials [51]. However, few studies have brought to bear the

superior temporal and spatial resolutions of MEG to elucidate the abnormalities in brain activity associated with movement activation and inhibition in PD.

Our goal was to examine PD-associated abnormalities in the magnitude and timing of oscillatory power in the basal ganglia thalamocortical circuit subserving response activation and inhibition. We tested the hypothesis that oscillatory activity in the frontal cortical regions subserving these functions had reduced event related power change and delayed onset in people with mild to moderate PD on dopamine replacement therapy compared to control participants. This hypothesis was based on previous work showing increased cortical resting state oscillatory synchronization and increased event related response latency in PD [35–40, 43, 52–54]. We used MEG, which combines high temporal precision and spatial resolution in the cortex to measure anatomical, at the level of the cortical field, region of interest, and physiological, specifically electrophysiological data. Unlike previous work, we matched motor output across trial types, allowing us to isolate the pre-movement processes of activation and inhibition to examine network dysfunction in PD.

## Methods

### Subjects

Individuals with PD were recruited from the movement disorders clinic at the UC Davis Medical Center. Clinical diagnosis of Parkinson disease was made according to Queen Square Brain Bank criteria [55] by a movement disorders neurologist and confirmed by review of medical records. Eighteen people with PD (4 female, 14 male) and 18 controls (8 female, 10 male) participated in this study. All participants were right-handed and PD participants had right side (of body) dominant PD, defined by side of initial symptom onset. Statistical power analysis based on Beste and colleagues [56] showed an amplitude difference between PD and control groups of about 8 ±2 μV in P300 peak amplitude for both compatible and incompatible Go trials. Using an alpha of 0.05 for a single comparison, a sample size of 18 per group yielded power of over 90% for an independent, 2 sample t-test comparing PD and control groups. Similar results were obtained from a power analysis of peak latency measures from the same study.

Inclusion criteria were male or female > 55 years old and fluent in English. Additional PD specific inclusion criteria were right-handed and right sided PD dominant onset. Exclusion criteria were history of severe head trauma based on patient report; significant uncorrected visual impairment; significant additional medical conditions known to affect cognitive function; history of substance or alcohol abuse; inability to understand informed consent, study purpose and procedures or other study materials involved in the research study; and women who were or might have been pregnant. Additional exclusion criteria included depression (Beck Depression Scale score >20; [57]), significant cognitive impairment (Mini-Mental State Exam <25; [58]), and excessive daytime sleepiness (Epworth Sleepiness Scale score >10; [59]). MRI specific exclusion criteria were significant claustrophobia or other identified problem making the MRI environment intolerable; body weight >300lbs; pacemakers, artificial limbs, or other implanted medical devices that contained metal; other metal objects, such as jewelry, piercings, braces, or internal prosthetics that were not MRI compatible and could not be removed based on pre-MRI screening form.

PD specific exclusion criteria were atypical PD, persistent tremor (score > 1 on UPDRS items 16, 20 or 21 for best on medication state), presence of motor fluctuations (score >1 on UPDRS items 36–39), or dyskinesia (score >1 on UPDRS items 32–34). Participants with PD were given the UPDRS on medication on a separate day from brain imaging. Written consent from each participant was obtained prior to the experiment and procedures were approved by

the Institutional Review Board on Human Subjects Research at the University of California, Davis.

All patients had late onset (>55 years at time of diagnosis) idiopathic PD with a history of positive response to dopamine replacement therapy and no alterations in medication for 6 weeks prior to enrollment. PD participants took their prescribed medication in the morning prior to study participation and performed the experiment in their best ON medication state. Thus, PD participants performed the tasks while on levodopa with carbidopa, dopamine agonist, Amantadine, Monoamine Oxidase Type B inhibitors and/or Catechol-O-Methyl Transferase inhibitor treatment for Parkinson disease. Dopamine equivalency for each PD participant's daily medications was calculated based on established conversion factors [60].

### Activation inhibition task

Subjects were presented with a visual cue-target design task and required to respond to the target by pushing a button with one or both index fingers. Fiber optic button boxes (Photon Control, Inc. <http://www.photoncontrol.com>) were held in both the left and right hands. Subjects were trained prior to entering the scanner to respond only to the target arrow(s) with their index fingers pressing hand-held button boxes. The training task consisted of a series of stimuli that contained 2 trials of each trial type. All subjects executed this trial run once before scanning following which they reported understanding the task. Stimuli were generated on a PC using Presentation software (www.neurobs.com/presentation). Stimuli were projected into the magnetically shielded room using a Christie Lx41 projector (Christine Digital, Cypress) onto a screen using a series of mirrors. Participants lay supine and their heads were padded to reduce movement.

On each trial, stimuli consisted of a visual fixation cross (a white + sign on a black background), which was present for the entire trial, and subjects were instructed to focus on this fixation point. A cue arrow appeared superimposed on the fixation cross, followed by a target arrow indicating the response to produce on a particular trial. The arrow cue (presented in orange) appeared for 200 ms followed by a delay interval that varied randomly between 600 and 1200 ms. The target arrow then appeared in the same central position (in blue) for 1000 ms. The inter-trial interval (ITI) was 2000–3000 ms (Fig 1A).

Cue and target stimuli were pairs of either unidirectional arrows pointing to the left or right, or bidirectional (double-headed) arrows. Subjects were instructed to respond to the target arrow with either a unimanual (in the direction of a single-headed arrow) or bimanual (in the case of bidirectional arrows) button press. One of seven possible pairs of cue/target stimuli was presented: bilateral cue and bilateral target (cue = B, target = B), right cue and bilateral target (RB), left cue and bilateral target (LB), right cue and right target (RR), left cue and left target (LL), bilateral cue and right target (BR), or bilateral cue and left target (BL). These combinations were grouped into two trial types: matched trials, where the cue and target arrows were the same (RR, LL, and BB), and mismatched trials, where the cue and target arrows were mismatched (BR, BL, RB, and LB). The mismatched trials were further classified into two key types that represent distinct types of behavioral functions. In the first type, subjects were required to activate an un-cued button press movement, as in the case of RB and LB trials. For these trials, a unilateral cue followed by a bilateral target requires initiation of a response that is un-cued for one hand, the hand not indicated by the cue. This added response (not previously cued) requires movement activation of a non-activated/non-cued response. In the second type, which we called inhibition trials, subjects were required to inhibit a cued movement, as in the BR and BL trials. In this case, a bilateral response is cued, but subjects must inhibit the already cued response on the target trial, responding with a single hand. In

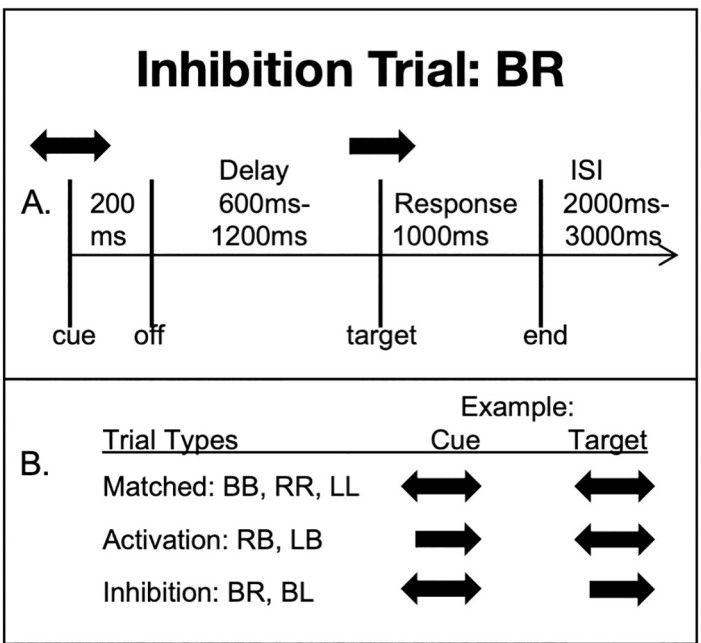

**Fig 1. Trial timeline.** (A) Arrow cue timeline for a bilateral cue and unilateral target. (B) Example of arrow cue visual presentation to subjects. The examples are for bilateral or right-hand trials. Left-hand stimulus pattern was identical. Letters indicate cue and target type, B = bilateral, L = left, R = right.

each of the 7 trial types there were 120 trials during a 70-minute scan session (some examples of trial types are shown in Fig 1B). Trial types were presented in random order.

## Behavioral data

Behavioral measures (reaction time and errors) were collected using ADC channels from voltage changes in the button box and extracted using CTF Data Editor software (https://www.ctf.com/products). Datasets were sorted by trial type (matched, mismatched: activation, mismatched: inhibition), and errors were detected by visually inspecting voltage changes in ADC channels. An error was defined as a unilateral button press of the incorrect button, a unilateral response when a bilateral response was required or a bilateral response when a unilateral response was required. Error trials were excluded from analysis. For epoching of data, target stimulus presentation was marked as 0 ms. Reaction time was calculated for correct trials as the time between the onset of the target arrow to the onset of the button press for each response.

## Acquisition

Neuromagnetic activity was recorded in a magnetically shielded room using a whole-head bio-magnetometer (CTF MEG, Coquitlam, Canada) with 275 first-order axial gradiometers and 27 reference sensors that enabled collection of synthetic third-order gradient data with improved signal-to-noise ratio. Coils at the nasion and 1 cm from the tragus, rostral to the left and right periauricular points in the direction of the nasion, were used to quantify head position relative to the sensor array. These points were later co-registered to the structural MRI through a multi-sphere head model. Scan sessions where head movement exceeded 5 mm within a run

were discarded and repeated. Epochs of 3–6 seconds duration were acquired using a sampling rate of 1200 Hz.

For reconstructions of the MEG data in source space, a structural MRI scan was acquired on a 1.5 T GE Signa scanner using an MP-RAGE (multiplanar rapidly acquired gradient echo) imaging sequence with the following parameters: repetition time (TR), 7.87 s; echo time (TE), 2.69 ms; flip angle, 8 degrees; slices, 200; field of view, 256 mm; resolution, 1 x 1 x 1.2 mm.

Data for this study has been made publicly available under Creative Commons CC0 1.0 Universal (CC0 1.0) at https://doi.org/10.12751/g-node.nb35ss. The data is stored in BIDS format using the CTF *.ds filetype. It can be opened in either proprietary CTF software (Omega 2000) or open-source M/EEG software packages (NUTMEG, fieldtrip). The Fieldtrip website expands on the file format here: https://www.fieldtriptoolbox.org/getting_started/ctf/.

## Analysis

Trials were corrected for noise and movement artifacts and error trials were discarded using DataEditor software (https://www.ctf.com/products). Noise and artifacts were identified visually by scanning trials for patterns caused by eye blinks, saccades, and head motion (MEG sensor amplitude exceeding 20 pT). Neural sources were estimated in the time-frequency domain using the Neurodynamic Utility Toolbox for MEG (NUTMEG; https://www.nitrc.org/projects/nutmeg/) across a shared computing cluster at the California Institute for Quantitative Biomedical Research (www.qb3.org). Changes in induced (non-phase locked) activity were estimated using an adaptive spatial filtering technique (lead field resolution = 5 mm), which effectively weights each source estimate (voxel) relative to the signal of the MEG sensors [61–63].

Source power for each voxel was determined by comparing the magnitude of the signal during an 'active' experimental time window versus a pre-stimulus baseline 'control' window [61, 64] using a noise-corrected pseudo-F statistic expressed in logarithmic units (decibels (dB); [61]). The data were stimulus-locked (target onset = 0 ms) followed by 25 ms time windows until 1112.5 ms post-target onset. Data were passed through a filter bank and partitioned into partially overlapping time windows of varying size (100, 150, 200, and 300 ms) optimized to capture spectral peaks in the MEG signal [61, 63, 65]. Twenty-five ms steps were then estimated in the alpha (8–12 Hz), beta (12–30 Hz), and gamma (30–55 Hz) bands.

Whole-brain reconstructions of oscillatory activity were co-registered with each individual's structural MRI. Each reconstruction was spatially normalized by applying a transformation matrix derived from the normalization of the structural MRI to a standard T1 template brain (Montreal Neurological Institute; MNI305) using SPM2 (http://www.fil.ion.ucl.ac.uk/spm/software/spm2). Averages and variance maps were smoothed using a Gaussian kernel with a 20 mm$^3$ width (full-width half-maximum) [63, 65]. Spatially normalized activation maps were entered into a group analysis using statistical non-parametric mapping (SnPM) [66], a statistical metric known to accommodate non-normally distributed MEG datasets [67]. Regions of interest (ROI) based on Brodmann nomenclature were derived from MNI coordinates in the normalized brain [68]. Permutation testing ($2^N$ possible combinations of negations) to assess significance were performed both within-group (one-sample t-test, mean difference between conditions) and between the control and PD groups (unpaired t-tests). Within group analyses were corrected for multiple comparisons at a familywise error rate cutoff of $p<0.05$. Between-group analysis was corrected with a false discovery rate threshold of $p<0.05$. For the gamma band, we began with a conservative (multiple comparison corrected) threshold of $p<0.05$ as a first step, with a more liberal threshold (significant at $p<0.05$, uncorrected) as a second step if significant effects were not observed at a conservative level. The details of this approach have

**Table 1. Demographic data reported as mean and (SD) except for H&Y, which is the median.** M = male, F = female, UPDRS = Unified Parkinson Disease Rating Scale.

| | N | Age (years) | Sex | Education (years) | UPDRS Total | UPDRS III | H&Y | Dopamine Equivalent (mg) | Disease Duration (years) | Age at Diagnosis (years) |
|---|---|---|---|---|---|---|---|---|---|---|
| **PD** | 18 | 66.5 (8.0) | 14M, 4F | 15.4 (2.7) | 32.7 (13.5) | 20.4 (12.0) | 2 | 462.5 (395) | 4.8 (2.7) | 61.6 (8.1) |
| **Control** | 18 | 63.8 (8.7) | 10M, 8F | 14.0 (2.3) | na | na | na | na | na | na |

been described elsewhere [61, 67, 69]. As we have noted previously [61], the reduced power inherent to higher frequency oscillations often require us to rely on more liberal thresholds with a concomitant increase in risk for type 1 error [61, 65].

Latency data was obtained from simple activation and inhibition trials (not contrasts). Onset and duration for each region of interest was determined by using the time frequency viewer to select the first and last 25 ms time window where the ROI was significant at its threshold. Between group differences in behavioral and latency data were evaluated using repeated measures analysis of variance (ANOVA) with post hoc analysis using a p value of 0.05.

## Results

### Subjects

There were no significant differences across groups for age $F(1, 35) = 0.65$, $p = 0.43$ or years of education $F(1,35) = 1.64$, $p = 0.21$. Gender distribution was not significantly different across groups $X^2(1, 35) = 2$, $p = 0.16$. Demographic data are presented in Table 1.

### Behavioral data

Repeated measures ANOVA revealed that there were no significant differences in reaction times for either group or any trial type $F(1,70) = 2.25$, $p = 0.14$. Error rates were significantly higher in the PD group for all trials $F(1,70) = 6.22$, $p = 0.015$. The means and standard deviations for reaction times and error rate are reported in Tables 2 and 3.

### Task based activity

In general we observed brain regions that showed statistically significant event related changes in oscillatory power (p<0.05, corrected for multiple comparisons at a familywise error rate;

**Table 2. Reaction time (ms) for control, activation, and inhibition trials.** Values are mean reaction time (SD) in milliseconds (ms) from target presentation.

| | Matched (ms) | Mismatched: Activation (ms) | Mismatched: Inhibition (ms) |
|---|---|---|---|
| CO | 533.95 (104.12) | 579.05 (99.38) | 531.03 (84.58) |
| PD | 570.60 (149.01) | 599.21 (163.67) | 586.06 (125.74) |

**Table 3. Error rate for control, activation, and inhibition trials.** Values are mean percentages (SD), calculated by number of incorrect trials / number of total trials. *p < .05.

| | Matched* | Mismatched: Activation* | Mismatched: Inhibition* |
|---|---|---|---|
| CO | 1.35% (3.23%) | 1.72% (4.66%) | 1.31% (3.16%) |
| PD | 4.4%(7.55%) | 3.7% (7.24%) | 4.3% (7.92%) |

not shown) following response (button press) activation or inhibition that included Brodmann area 4 (motor cortex), lateral BA 6 (pre-supplementary motor area), medial BA 6 (supplementary motor area), BA 7 (posterior parietal cortex), BA 24 (anterior cingulate cortex), and BA 9/10 (medial and anterior prefrontal cortex, specifically medial and superior frontal gyrus). Areas 9 and 10 were reported together because activity frequently overlapped at the border between these two regions. Right- and left-hand trials both tested activation and inhibition in a similar fashion, and results from right- and left-hand trials were similar, so data from right-hand response trials were used to illustrate the results. Right-hand results were chosen because all subjects were right-handed with right side dominant disease. Right- and left-hand response MEG data were not combined because frontal and sensorimotor cortex results were not aligned due to crossed inputs from the two hands, while frontal lobe activation was independent of response hand.

Across all tasks (control subject matched trial data not shown), primary motor cortex, lateral pre-supplementary motor area, and posterior parietal cortex showed statistically significant decreases in power compared to baseline, while supplementary motor area and anterior cingulate cortex showed increases. In medial/ anterior prefrontal cortex, we observed both increases and decreases in power in multiple frequency bands and time points. Of interest here are results from 1) control subject motor planning (matched vs. mismatched trial contrasts for activation (Fig 2A) and inhibition (Fig 4A); and 2) difference in motor planning across disease groups (control vs. PD subject mismatched trials contrasts for activation (Fig 2B) and inhibition (Fig 4B). Significant PD associated power changes in specific frequency bands, latencies, and brain regions are described below.

## Activation contrast analysis (unilateral cue, bilateral target vs. bilateral cue, bilateral target)

For activation trials, subjects were presented with a unilateral cue and bilateral target; for example, right cue, bilateral target (RB), which required activation of a (un-cued) left-hand response. In control subjects, brain activity from these trials was compared to that from control trials matched for motor output that did not require response activation (BB, bilateral cue, bilateral target; Fig 2). The control group contrast analysis (Fig 2, left, false discovery rate corrected threshold of p<0.05) revealed peak increased power in the gamma band in left medial/ anterior prefrontal cortex at 337.5 ms. There was also a peak power increase in the alpha band in anterior cingulate cortex at 337.5 ms. In bilateral medial/ anterior prefrontal cortex, there were peaks in increased power in the beta band at 537.5 ms.

To identify differences in PD brain activity patterns following response activation (of a not previously cued response), we compared trials consisting of a unilateral cue and bilateral target in the control versus PD groups. The contrast analysis of mismatched activation trials in the PD group versus control group (RB PD vs. RB Control; Fig 2, right; false discovery rate corrected threshold of p<0.05) revealed a relative power increase in the beta band in the bilateral primary motor cortex that peaked at 387.5 ms. We identified a decrease in power in the alpha band in left supplementary motor area, a decrease in power in the beta band in the left medial/ anterior prefrontal cortex, both at 387.5 ms. In the beta band we also observed a decrease in power in posterior parietal cortex at 512.5 ms in the left hemisphere in the PD versus the control groups.

Repeated measures ANOVA of response latency data from both matched and mismatched (activation and inhibition) trials revealed that changes in power peaked at longer latencies in the PD group relative to those in the control group. For matched (BB) trials (Fig 3A), the PD group gamma band peak power change latency was delayed relative to controls in the left

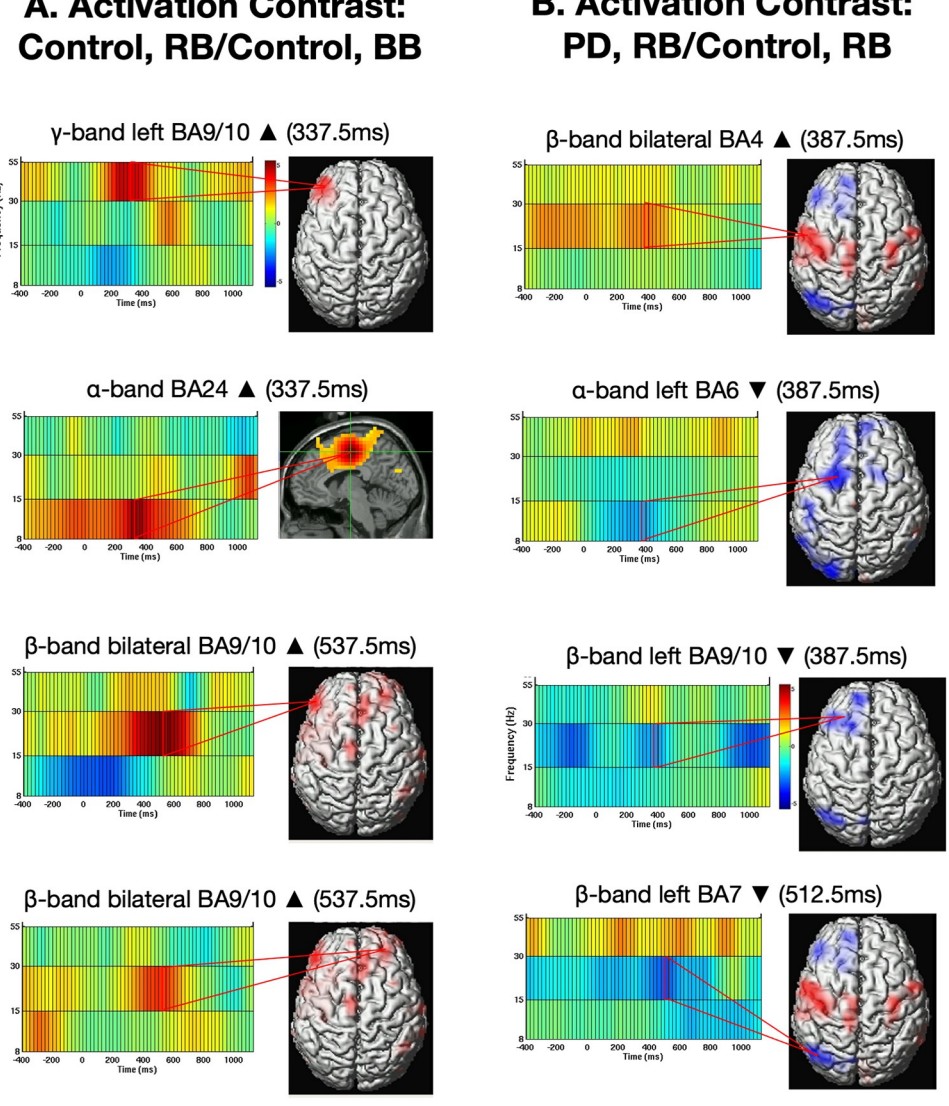

**Fig 2. Activation contrast analysis.** (A) Differences in peak intensity in the control group for mismatched activation trials versus matched control trials. (B) Differences in peak intensity during mismatched activation trials between the PD and control groups. MS = millisecond. Scale color bar represents power in arbitrary units. Peak activity indicates a significant change at p<0.05, corrected for multiple comparisons except for the gamma band. To identify gamma band activity, we used a more liberal threshold (significant at p<0.05, uncorrected). R = right hand, B = bilateral.

supplementary motor area (BA 6; t(34) = -2.595, p = 0.014), and in the beta band in left medial/ anterior prefrontal cortex (BA 9/10; t(34) = -2.201, p = 0.035). For mismatched activation trials (Fig 3B), peak changes in power also occurred at longer latencies for the PD group, but to differing degrees depending on the area, leading to a significant interaction between ROI and group (F(3.95, 134.26) = 4.504, p = 0.002). PD group peak latency was longer in the gamma band in left supplementary motor area (BA 6; t(34) = -2.595, p = 0.014) and in the beta band in left primary motor cortex (BA 4; t(34) = -2.377, p = 0.023). PD group peak latency was also longer in the alpha band in left medial/ anterior prefrontal cortex (BA 9/10; t(34) = 2.175, p = 0.037).

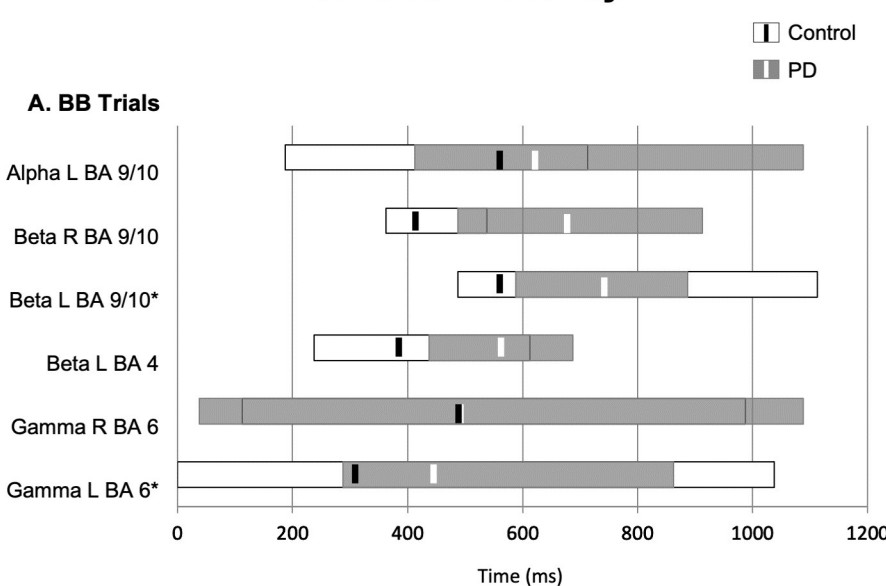

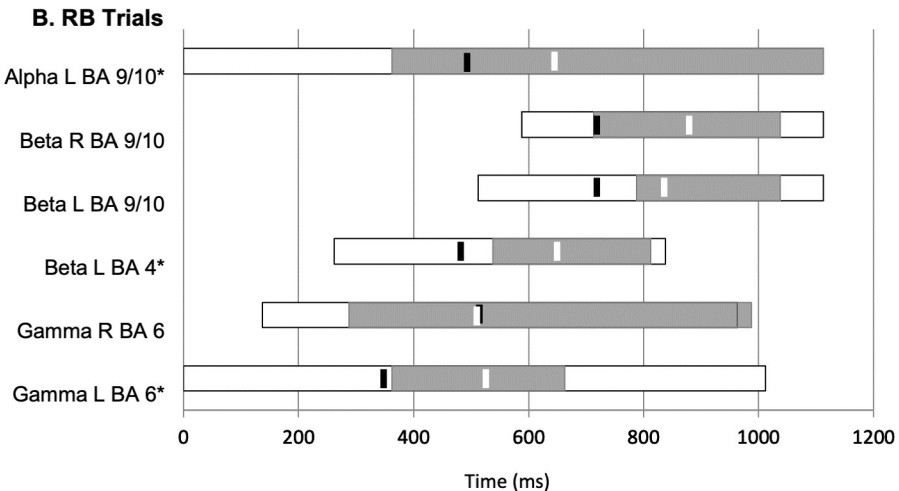

**Fig 3. Activation latency.** (A) Onset and duration (gray and white bars) of power change in each ROI for the matched control trials. Latency of peak power change is indicated by solid black or white lines. (B) Onset and duration of power change in each ROI for the mismatched activation condition. Peak latency was significantly longer in the PD vs. control group: *p < 0.05.

## Inhibition contrast analysis (bilateral cue, unilateral target vs. unilateral cue, unilateral target)

For response inhibition trials, subjects were presented with a bilateral cue and unilateral target; for example, bilateral cue, right target (BR), which required inhibition of a left-hand cued response. In control subjects, brain activity from these trials was compared to control trials matched for motor output that did not require response inhibition (RR, unilateral cue, unilateral target; Fig 4A).

Control group contrast analysis (Fig 4A) revealed statistically significant peak increased power (p<0.05, corrected for multiple comparisons at a familywise error rate) in the gamma

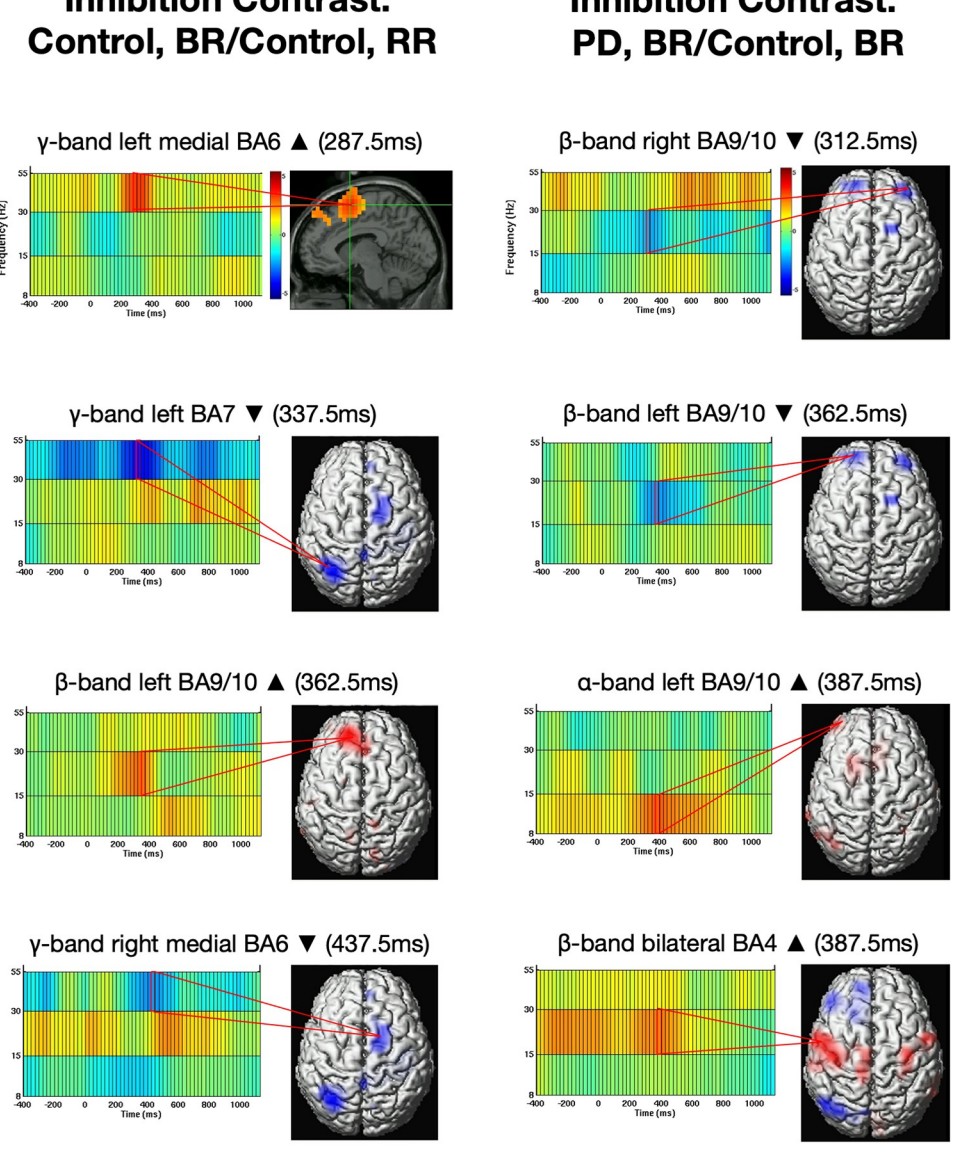

**Fig 4. Inhibition contrast analysis.** (A) Differences in peak power change in the control group for mismatched inhibition trials versus matched control trials. (B) Differences in peak power change during mismatched inhibition task between the PD and control groups. Conventions as in Fig 2.

band in medial supplementary motor area (BA 6) at 287.5 ms. Maximum decreased power was observed in the gamma band in left posterior parietal cortex (BA 7) at 337.5 ms, while peak increased power was found in the beta band in left medial/ anterior prefrontal cortex (BA 9/ 10) at 362.5 ms. We additionally observed maximum decreased power in the gamma band in right pre-supplementary motor area (BA 6) at 437.5 ms.

To identify differences in PD brain activity patterns following response inhibition, we compared control bilateral cue unilateral target trials to the same trials in the PD group. The contrast analysis of mismatched inhibition trials in the PD group versus the control group (Fig 4, right, false discovery rate corrected threshold of p<0.05) revealed bilateral decreases in beta power in medial/ anterior prefrontal cortex at 312.5 ms (maximum right hemisphere) and

# Inhibition Latency

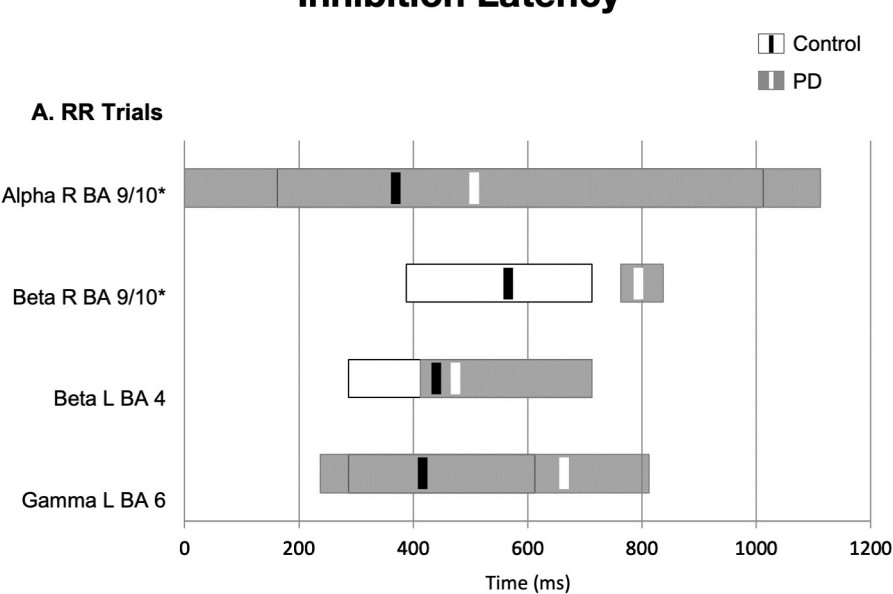

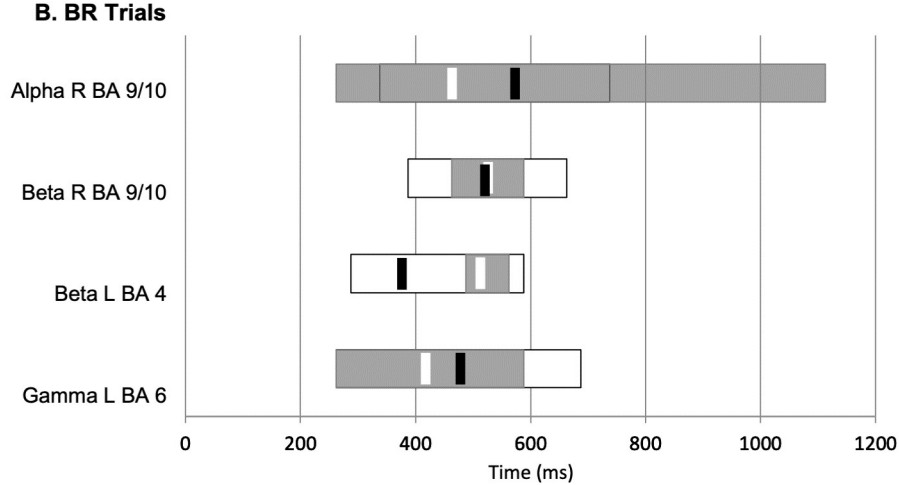

**Fig 5. Inhibition latency.** (A) Onset and duration (gray and white bars) of power change in each ROI for the matched control trials. Latency of peak intensity is indicated by solid black or white lines. (B) Onset and duration of power change in each ROI for the mismatched inhibition condition. Peak latency was significantly longer in the PD vs. control group: * p < 0.05.

362.5 ms (maximum left hemisphere) in PD that was not present in control data. Increased alpha band power was observed in left medial/ anterior prefrontal cortex in the PD compared to the control group with a peak latency of 387.5 ms. We saw increased power in PD compared to controls in the beta band in primary motor cortex with a peak latency of 387.5 ms. We also observed greater power decreases in the gamma band in the PD group compared to controls with peak latencies of 112.5 and 612.5 ms in contralateral pre-supplementary motor area, indicating delayed onset and termination of desynchronous activity in this region in the PD group (Fig 5B).

Repeated measures ANOVA of response latency data from both matched and mismatched trials revealed that changes in power peaked at a longer latencies in the PD group relative to

those in the control group. We observed differences in peak latency between the PD group and the control group only for matched trials (RR; Fig 5A). PD group peak latency was longer in right medial/ anterior prefrontal cortex in the alpha band (t(34) = -3.213, p = 0.003) and in the beta band (t(34) = -3.327, p = 0.002). Variability was high in gamma latency in left lateral pre-supplementary motor area. There were no peak latency differences between groups in the mismatched inhibition trials (BR; Fig 5B).

## Discussion

We tested the hypothesis that oscillatory activity in the frontal cortical regions underlying response activation and inhibition had reduced event related power change and delayed onset in PD compared to control participants. We extend previous work by providing cortical oscillatory power and latency data with ROI spatial resolution specific to motor planning, and describe abnormalities in the cortical activation and inhibition responses associated with PD. Despite similar reaction time performance, in PD motor cortex we found increased beta band synchronization and delayed peak latency in primary motor cortex as in previous studies (see [46] for review). We also found reduced event related power changes on correct trials throughout the frontal cortical regions subserving response activation and inhibition consistent with previous work (for review see [34, 70]). Peak latency was delayed in PD for movement activation across frequency bands and brain regions while inhibition data was not different across groups. Furthermore, the changes in location and oscillatory power change of response were consistent with the concept of a compensatory mechanism.

### Anatomy and physiology of movement activation and inhibition in PD

We found a group of cortical regions subserving movement activation and inhibition that included prefrontal, cingulate and supplementary motor cortex. In the control group we found increased beta band power in the left medial frontal gyrus during inhibition of a cued movement. Left medial prefrontal cortex is thought to be involved in sustained attention [71–73], short term memory [74], perceptual decision making [75], and overriding automatic responses [76, 77], all of which are likely components of our task. In PD we observed that activation trials were associated with reduced alpha band power changes and delayed peak latency in prefrontal cortex. This region plays a role in the parallel inhibitory and excitatory regulation of neural activities associated with executive function (see [34] and [78] for a review) which are known to be disrupted in PD [79]. Furthermore, the left prefrontal cortex [80] and BA9 subserve error detection [76, 81, 82], and the inhibition response was attenuated in this region in people with PD. Damage to dorsomedial prefrontal cortex (anterior cingulate cortex and supplemental motor area) has also been linked to increased errors in Go Nogo paradigms [18]. Taken together, these findings suggest that increased error rate in PD may be related to abnormal activity in prefrontal cortex.

We also observed event related power changes in cingulate and supplementary motor cortex during movement activation and inhibition that appeared to be preserved in PD. The anterior cingulate, specifically the dorsal anterior cingulate or midcingulate cortex [83, 84], is functionally linked with the prefrontal cortex [85–87], and is likely to play a role in attention processing [53, 88], working memory [76, 89, 90], conflict [91–95], and error processing [96, 97]. While we found that event related alpha band power changes in the cingulate were preserved in PD, previous data from this region showed significantly higher levels of slow wave activity during resting conditions in people with PD [53, 54]. In fact, while Braak staging suggests that anterior cingulate involvement occurs in later stage PD [98], there is accumulating evidence of abnormalities in cortical thickness, cerebral blood flow, fractional anisotropy,

dopamine-2 receptor binding and connectivity earlier in the disease (for review see [99]). Our contrasting findings may be related to the significant difference in experimental design and measurement modality across studies and to the impact of plastic compensatory brain changes [100] on functional measures like ours.

In medial supplementary motor area in the control group we observed increased gamma band power during inhibition of a cued movement that was preserved in PD. The role of the right pre-supplementary motor area in response inhibition has been well documented (e.g., [18, 81, 101–106]), and it has been established that the pre-supplementary motor area is involved in motor response inhibition [19, 97, 104, 107, 108], active during stop signal and 'change of plan' task paradigms [105] (see [109] for a review), and may be associated with decision making [110], mediation of attention and motor response activation [111]. The normal gamma band power changes may be related to the relatively healthy inhibition related peak power changes observed in PD. In contrast, we found decreased power and increased latency in the alpha band for activation in medial supplementary motor area in PD compared to controls. The link between supplementary motor area function and timing of motor planning such as anticipatory postural adjustments is impaired in PD [112]. It is interesting to note that motor responses were required for both activation and inhibition trials, the difference being that activation requires execution of an un-cued movement. Behavioral deficits specific to un-cued, as opposed to cued movement initiation are well documented in PD [12, 113–115], which is consistent with our data showing delayed peak latency in motor premotor and supplementary motor cortex for activation trials.

## Frequency specific cortical oscillatory power

Magnetic and electrical signal synchronization and desynchronization are thought to reflect dynamic communication between spatially distributed brain regions [116, 117]. Oscillatory power at different frequencies has been associated with unique behavioral functions (for a review, see [118]). Power in the gamma band (33–60 Hz) reflects active engagement (i.e., in feature integration, attention, and movement preparation, depending on the cortical area) [119, 120]. In controls, changes in gamma band power for activation and inhibition networks [93, 121–124]. The pattern of gamma band activity was similar to the peripheral attention network described by Corbetta and colleagues [125], and appeared to be relatively intact in PD. We found no differences in gamma band power change amplitude or location in PD. However peak latency was delayed in several ROI's, consistent with existing studies showing reduced speed of processing in PD [122, 126–128]. In contrast, inhibition related gamma band changes have been reported in the subthalamic nucleus in PD [129], particularly in response to failed inhibition trials [130]. and increased gamma desynchronization following a Go cue was positively correlated with reaction time in PD [130]. Cortical gamma activity is also modulated by thalamic alpha activity [131], and Yoo and colleagues [132] found that alpha-gamma coupling was higher in PD compared to controls.

Reduced power in the beta band frequencies has been associated with movement preparation and execution [7, 133–135] as well as cognitive control [118]. Changes in beta power prior to target onset have also been shown to correlate with reaction time in PD and healthy controls [133]. Furthermore, healthy adults show an increase in beta band power over the frontal cortex while inhibiting responses during a stop-signal task [101]. As in previous work [22, 133, 136, 137], changes in beta band power were common in PD. Reduced beta band desynchronization in motor and prefrontal cortex during activation and inhibition may be related to reported increased synchronization in this region at rest in PD [43]. Interestingly, Heideman and colleagues [135] examined PD associated power differences in beta band, differentiating event

amplitude, duration, and interval time. They found that beta-state interval time between short-lived, high-amplitude events accounted for decreases in beta band power in averaged responses in PD.

Alpha band activity is increased in cortical areas not engaged in a task [118], and event related synchronization in the alpha band is associated with top down control of response inhibition [138]. Previous work has shown that power changes in the alpha band prior to target onset, which were attenuated in PD, were correlated with reaction time [139, 140]. Across frequencies, where control subjects showed decreased event related power, this decrease was attenuated in PD. For example, we found decreased alpha band power in PD compared to controls in the supplementary motor area and medial/ anterior prefrontal cortex for response activation and inhibition, respectively. Decreased power in the alpha band in the frontal lobe has been associated with executive dysfunction in PD [141]. As in our study, Perfetti and colleagues [139] showed decreased alpha power in fronto-parietal cortex in PD prior to the presentation of a target. The change in alpha band power from baseline was positively correlated with reaction time in a reaching task [139]. Again, the attenuated event related changes in power in PD may be related to increased synchronization in resting state alpha band activity [141].

## Compensation

There is a well described lag between the onset of nigrostriatal nerve terminal degeneration and the onset of motor signs in PD [142, 143] that is consistent with a capacity to compensate. In our study, reaction time did not differ significantly between groups, while differences in power were apparent in the network subserving activation of a cued movement, indicating that behavior was preserved in the face of changing brain function. For example, the decrease in alpha power in supplementary motor area during PD activation could be interpreted as recruitment of this region as a compensatory mechanism. Similarly, Buhmann and colleagues [144, 145] observed differences in left dorsal premotor cortex fMRI activity between asymptomatic Parkin mutation carriers and healthy non-carriers who performed a finger to thumb opposition task. Changes in left lateral supplementary motor area were only observed in response to a reduced cue condition where subjects were required to select which finger to tap. In addition, compensatory changes in connectivity in this region have been reported [145]. Compensation, therefore, is a possible explanation for the changes in alpha activity we observed in supplementary motor area [145].

However, disentangling compensation from pathophysiology is difficult. The simultaneous abnormal peaks (387.5 ms) in medial prefrontal cortex, primary motor cortex and supplementary motor area in the alpha and beta bands in PD activation could result from a single widely connected pathological source of circuit dysfunction. For example, the subthalamic nucleus, which is known to be dysfunctional in PD [146], has shown power changes during both movement activation and inhibition in PD [129, 147, 148]. Conversely, the additional distributed activity in primary motor cortex and supplementary motor area could be related to compensation for failing dopaminergic innervation, which is also consistent with the relatively normal response activation reaction times in the PD group. In a recent review of MEG and PD studies, Boon and colleagues [46] report that dopamine replacement therapy or deep brain stimulation normalized beta band power and interregional coupling while alleviating motor symptoms. These authors [46] suggested that the increased beta band power and connectivity in PD was a compensatory mechanism which became redundant once dopamine replacement therapy was administered.

Another possible interpretation of the abnormal distribution of activity is a change is response strategy. While reaction time was similar across groups, errors were more frequent in

the PD group, suggestive of a shift in speed/accuracy trade off, maintaining speed at the cost of accuracy. This adjustment may be related to the recruitment of additional brain areas as well. Thus, while our findings of additional cortical power changes in PD are consistent with circuit dysfunction involving premotor and prefrontal cortex, the lack of reaction time deficits or changes in the speed/accuracy trade off in conjunction with activity observed in regions not commonly associated with PD pathology, such as posterior parietal cortex, could represent compensatory activity, though again, it is difficult to definitively differentiate compensation and pathological disinhibition [149, 150].

## Clinical relevance

Currently, the utility of MEG in clinical practice is limited [46, 153]. The clinical significance of common MEG findings, such as the link between whole brain resting state spectral slowing and cognitive impairment are not clear. There is also a lack of consistent and comprehensive data examining the impact of dopamine replacement therapy and DBS on MEG outputs in PD ([46] for review). However, in development of clinical treatment advances which include forms of neurostimulation, understanding the underlying neural dynamics is crucial. Our findings inform those aspects of understanding. For example, there may be clinical relevance to hypothesized association of changes in alpha-band activity in supplementary motor cortex with compensatory mechanisms [144, 145]. may be possible to facilitate the brain's intact compensatory mechanisms using neuromodulatory interventions that boost alpha power in BA 6 [151, 152]. However the clinical implications of the current results, though potentially significant, remain largely a matter of speculation.

Our observation of more widespread abnormalities in the neural activity associated with movement activation, as compared to inhibition, is consistent with the general characterization of parkinsonism as, first and foremost, a disorder of movement and with the preferential vulnerability of dopaminergic innervation of the basal ganglia's motor territories [153, 154]. Indeed, many of the classical cardinal signs of PD (e.g., akinesia, bradykinesia, postural instability) fall easily within the general category of impairments of motor activation [155]. Data on abnormal cortical activity may inform intervention strategies [156–158]. For example, cognitive neurorehabilitation targeting movement activation has shown some success [157, 158]. Future studies of neurorehabilitation using MEG could yield clinically significant insight into the plasticity subserving recovery of function. Impairments of movement inhibition, though clearly present in PD, were described only recently [2] and, arguably, make subtle and complex contributions to classic motor signs such as gait or intricate stepping abnormalities [156]. As in most complicated diseases, the picture of Parkinson's disease becomes clearer with accumulation of evidence at all levels of the neural-cognitive system.

## Limitations

Our ability to measure magnetic task based responses was limited to the cortex, specifically to tangential signals emanating from sulci [159, 160], because of the physics of MEG. Our sensors did not capture signal from subcortical structures such as the basal ganglia which clearly play an important role in response activation and inhibition [11, 27, 161, 162]. We did not evaluate the impact of dopamine replacement therapy on movement activation and inhibition though this response is significant [163, 164] (for review, see [165]). Our analysis of the gamma frequency band was conducted using an uncorrected p value. Despite our best efforts to reduce noise, for example enrolling only right-handed subjects with right side disease onset, gamma band activity was noisy, and the more stringent corrected p value yielded minimal consistent activation patterns. Thus our findings on gamma band activity have an increased susceptibility

to Type 1 error, though no differences between PD and control participants were observed in the gamma band event related power changes. We did find changes in the location, amplitude, frequency and latency of response signals in optimally medicated PD participants. However disentangling the contribution of PD pathophysiology, network compensation and chronic, as opposed to the more natural event related sporadic fluctuations in dopamine is a daunting task that is key to understanding the disease mechanisms subserving PD.

## Conclusions

Our findings are consistent with decreased event related power changes in the frontal cortex during movement activation and inhibition in PD. However the PD associated changes in the network subserving movement activation were widespread across frontal cortex and included motor regions, while those for inhibition were not as pronounced, though both trial types required a motor response. In addition, PD associated increases in peak latency were observed only in movement activation data, suggesting that deficits in movement production were more complex and potentially influential than deficits in inhibition. While difficult to dissociate from disease pathophysiology and medication effects, the changes in location and power of response were consistent with the concept of compensation following cell death in the substantia nigra.

## Acknowledgments

We would like to thank Susanne Honma for assistance with data collection and analysis. Dr. Karen Sigvardt passed away before the submission of the final version of this manuscript. Dr. Elizabeth A. Disbrow accepts responsibility for the integrity and validity of the data collected and analyzed.

## Author Contributions

**Conceptualization:** Elizabeth A. Disbrow, Elizabeth A. Franz, Robert S. Turner, Karen A. Sigvardt.

**Data curation:** Elizabeth A. Disbrow.

**Formal analysis:** Elizabeth A. Disbrow, Elizabeth M. Dressler, Kimberley Russo, Maria I. Ventura, Leighton Hinkley.

**Funding acquisition:** Elizabeth A. Disbrow, Karen A. Sigvardt.

**Investigation:** Elizabeth A. Disbrow, Elizabeth M. Dressler, Kimberley Russo, Maria I. Ventura.

**Methodology:** Elizabeth A. Disbrow, Leighton Hinkley.

**Project administration:** Elizabeth A. Disbrow.

**Resources:** Elizabeth A. Disbrow.

**Software:** Leighton Hinkley, Srikantan S. Nagarajan.

**Supervision:** Elizabeth A. Disbrow.

**Validation:** Elizabeth A. Disbrow.

**Visualization:** Elizabeth A. Disbrow.

**Writing – original draft:** Elizabeth A. Disbrow, Nathaniel D. Glassy, Elizabeth A. Franz, Robert S. Turner, Leighton Hinkley.

**Writing – review & editing:** Elizabeth A. Disbrow, Nathaniel D. Glassy, Elizabeth A. Franz, Robert S. Turner, Leighton Hinkley, Richard Zweig, Christina R. Ledbetter.

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
