## [Decision Letter · Decision Letter 0]

22 Sep 2020

PONE-D-20-21985

Cortical Oscillatory Dysfunction in Early Parkinson Disease During Movement Activation and Inhibition

PLOS ONE

Dear Dr. Disbrow,

Thank you for submitting your manuscript to PLOS ONE. After careful consideration, we feel that it has merit but does not fully meet PLOS ONE’s publication criteria as it currently stands. Therefore, we invite you to submit a revised version of the manuscript that addresses the points raised during the review process.

One reviewer rejected your manuscript, but two reviewers indicated major revisions. I have decided to give you the opportunity to address the suggestions and answer the questions. Please address the comments of three reviewers adequately.

We look forward to receiving your revised manuscript.

Kind regards,

Fabio A. Barbieri, PhD

Academic Editor

PLOS ONE

Journal Requirements:

"This work was supported by a grant to ED from the National Institute of Neurological Disorders and Stroke (RO1NS064040)."

"The funders had no role in study design, data collection and analysis, decision to publish, or preparation of the manuscript"

Reviewers' comments:

Reviewer's Responses to Questions

**Comments to the Author**

1. Is the manuscript technically sound, and do the data support the conclusions?

Reviewer #1: No

Reviewer #2: Yes

Reviewer #3: Yes

2. Has the statistical analysis been performed appropriately and rigorously? 

Reviewer #1: Yes

Reviewer #2: Yes

Reviewer #3: Yes

3. Have the authors made all data underlying the findings in their manuscript fully available?

Reviewer #1: No

Reviewer #2: Yes

Reviewer #3: No

4. Is the manuscript presented in an intelligible fashion and written in standard English?

Reviewer #1: No

Reviewer #2: Yes

Reviewer #3: Yes

5. Review Comments to the Author

Reviewer #1: This manuscript describes the effects on brain activity using MEG on response activation and inhibition in control and PD patients. The results show that different brain regions are impacted in control and PD. The study is purely correlative and does not provide a clear conceptual context to explain the results. The discussion is not well organized and difficult to read.

The manuscript is poorly written and very difficult to follow. The description of the tasks is confusing. For instance, in the methods, it says that in the RB and LB task, the subjects must press an un-cued button but the task seems to be cued because it is called RB and LB. The authors should clarify this because they use the word clue for BR and BL but they consider the task un-cued. In addition, there is no description in the methods on how the subjects enter their responses. It seems that they press a button with the right and/or left hand to choose their answers but this is not stated. In this context, results mention “right-hand trials” and left-hand trials” but the manuscript does not explain what this means. This is all the more confusing because on page 12 and 13 of the results, it is stated that “right- and left-hand trials were similar, so data from right-hand response trials were used to illustrate the results. Right-hand results were chosen because all subjects were right-handed with right side dominant disease. Right- and left-hand response MEG data were not combined because sensory and motor cortex results were not aligned due to crossed inputs from the two hands, while frontal lobe activation was independent of response hand”.

Reviewer #2: This is a very interesting study where the authors deeply discussed the cortical oscillatory dysfunction in early Parkinson's disease during movement activation and inhibition. I have some comments about this paper, and some suggestions too.

In general, the manuscript sounds very technical and scientific. However, some clinical statements were not clear. For example, in the title the authors declare that in this study, the participants were in early stage of PD. Can the authors please update the Hoehn and Yahr stages of which the participants were classified in their early stages?

I also suggest that the authors reduce the number of acronyms. Try to keep the acronyms to minimum.

A more detailed suggestion and review follows below:

Abstract

Page 2, line 48 – please, define SMA.

Introduction

Page 3, lines 53-54 – Please, give a reference to the statement made.

Page 3, line 60 – What motor signs of PD are you referring? Only akinesia, as cited in line 61? Please, be clearer about this statement.

Page 3, line 64 – Please state the “pre-movement external cueing” which are cited in the references 9-13. Such statement makes it easier for the reader.

Page 3, line 74 – I think the authors are referring to “Miller”, not colleagues here. If I am correct, please, change this.

Page 3, line 76 – Please, define BOLD.

Page 3, line 77 – Please, define RT.

Page 4, line 101 – Please, define UPDRS.

Methods

Subjects – Is this a powered sample, or 36 participants in total were chosen by convenience? Please make this statement.

Page 6, lines 120-122 – Please, cite the reference about clinical diagnosis of PD. In addition, were these participants diagnosed with idiopathic PD? Please, make it clear.

Page 6, line 127 – Please, define MRI.

Page 6, line 131 – Please, define MMSE.

Page 6, line 132 – Please, define ESS.

Page 6, line 133 – How did the authors define “severe tremor”? Did the authors use any tools for that? Please, make it clear.

Page 6, line 137 – Please, define COMT.

Page 9, line 193 – Please, define MR.

Page 10, line 234 – Instead of citing the references 47, 53, please state the details that justify the use of uncorrected multiple comparisons.

Page 10, line 235 – Please, define ROI.

Page 11, line 239 – Please, define ANOVA.

Discussion

Page 17, line 373 – Please, define ACC.

Page 17, line 386 – Please, define STN.

Page 18, line 410 – Please, define IFG.

Page 19, line 430 – Please, define PFC.

Page 21, line 470 – Please, define fMRI.

I missed 3 important keys in this manuscript: statement of limitations, clinical relevance and conclusion. In regards of limitations, can the authors define any limitation in this study? I understand the importance of these results to understand how PD affects movement control and inhibition, but can the authors work out the clinical relevance of these findings? Finally, a conclusion is indeed missing.

Table 1. Is there any statistical between groups for age and gender? Regardless of statistical difference, please state the p-values in the text, or table.

Did the authors use UPDRS total score? Please, present the part 3 scores as well. The authors mentioned in the discussion that no correlation was observed between UPDRS and cortical data. Have the authors observed any correlations using the motor score (part 3)? Why did the authors use UPDRS instead of MDS-UPDRS?

How was the dopamine equivalent calculated? Did authors use Levodopa equivalency daily dosage according to Tomlinson et al., 2010?

Reviewer #3: The authors' objectives were to examine PD-associated abnormalities in the magnitude and timing of oscillatory power in cortical networks subserving response activation and inhibition. The paper is interesting and contributes to the understanding of cortical networks subserving response activation and inhibition.

Major reviews:

What do the authors call early Parkinson disease? Table 1 shows that some patients have more than 8 years of illness.

The hypothesis needs further explanation.

How many attempts at practice have been made? Did PD participants and control have the same practice time? Could this have influenced the task's performance during testing?

Were patients evaluated ON or OFF? What does this interfere with the results?

Were the conditions randomized or presented in blocks? Was the task learned during the experiment?

Discuss the implications of having found differences in cortical activity despite the same task performance.

A Conclusion section would be important.

Minor review:

- Place the units in the tables

- what is the patients' H&Y?

- are the values in table 3 reported as mean and standard deviation?

- check the link to the page (http://www.sourcesignal.com/dataeditor.html) that is not working

- What was considered a wrong attempt?

6. PLOS authors have the option to publish the peer review history of their article (what does this mean?). If published, this will include your full peer review and any attached files.

Reviewer #1: No

Reviewer #2: No

Reviewer #3: **Yes: **Daniel Boari Coelho

---

## [Author Response · Author response to Decision Letter 0]

25 Jan 2021

We would like to thank the reviewers for the time and effort that they invested in reviewing this manuscript. All comments have been carefully considered and included in the revised manuscript.

Reviewer #1: 

We understand the frustration expressed by Reviewer 1 that the manuscript is very difficult to follow. The data set is complex, including several trial types and response options with both the right and left hand, as well as temporal and spatial brain response data across multiple frequency bands. It has been challenging to sift through everything and distill it into a reader friendly format that captures all the findings. I would like to thank the reviewer again for his/her feedback, which is invaluable in clarifying the manuscript. Based on the reviewer’s comments we have added a summary of the results to the first paragraph of the discussion to make the material easier to digest. We have also edited the manuscript throughout for clarity, including removing background information that was not directly related 

The description of the tasks is confusing. For instance, in the methods, it says that in the RB and LB task, the subjects must press an un-cued button but the task seems to be cued because it is called RB and LB. The authors should clarify this because they use the word clue for BR and BL but they consider the task un-cued.

The section describing the task, now called “Activation Inhibition Task” (starting on page 7, Line 157) has been revised to clarify the cued and un-cued/ activation vs. inhibition design. 

 In addition, there is no description in the methods on how the subjects enter their responses. It seems that they press a button with the right and/or left hand to choose their answers but this is not stated. In this context, results mention “right-hand trials” and left-hand trials” but the manuscript does not explain what this means. 

In the original version, this information was contained in the last sentences of the (lengthy) description of the task. It has been moved to the first few sentences of the revised version, along with a more general description of the task: “Subjects were presented with a visual cue-target design task and required to respond to the target by pushing a button with one or both index fingers. Fiber optic button boxes (Photon Control, Inc. <http://www.photoncontrol.com/>) were held in both the left and right hands. Subjects were trained prior to entering the scanner to respond only to the target arrow(s) with their index fingers pressing hand-held button boxes.” (page 7 line 158)

This is all the more confusing because on page 12 and 13 of the results, it is stated that “right- and left-hand trials were similar, so data from right-hand response trials were used to illustrate the results. Right-hand results were chosen because all subjects were right-handed with right side dominant disease. Right- and left-hand response MEG data were not combined because sensory and motor cortex results were not aligned due to crossed inputs from the two hands, while frontal lobe activation was independent of response hand”.

To clarify this procedure we added the phrase “Right- and left-hand trials both tested activation and inhibition in a similar fashion, and results from right- and left-hand trials were similar, so data from right-hand response trials were used to illustrate the results. Right-hand results were chosen because all subjects were right-handed with right side dominant disease. This change, in combination with the clarification of the task description, should improve reader accessibility to the information.

Reviewer #2: This is a very interesting study where the authors deeply discussed the cortical oscillatory dysfunction in early Parkinson's disease during movement activation and inhibition. I have some comments about this paper, and some suggestions too.

In general, the manuscript sounds very technical and scientific. However, some clinical statements were not clear. For example, in the title the authors declare that in this study, the participants were in early stage of PD. 

Can the authors please update the Hoehn and Yahr stages of which the participants were classified in their early stages?

The H & Y data has been added to table 1. In addition the word “early” has been removed from the title and the subject description.

I also suggest that the authors reduce the number of acronyms. Try to keep the acronyms to minimum.

We agree that there were too many acronyms. All acronyms were removed except for these relatively common ones:

ANOVA

EEG

fMRI

MRI

ROI

UPDRS

All acronyms are now spelled out the first time they are used.

Introduction

Page 3, lines 53-54 – Please, give a reference to the statement made.

On page 3, line 54 we have added references (Disbrow et al., 2013; Franz, 2006; Obeso et al., 2011) to the first sentence of the introduction for our statement “Response initiation and inhibition are functions fundamental to executive control that are disrupted in Parkinson disease (PD).” 

Page 3, line 60 – What motor signs of PD are you referring? Only akinesia, as cited in line 61? Please, be clearer about this statement.

We clarified this point by qualifying the original statement:

“Furthermore, impairments in response activation and inhibition have been proposed to subserve some of the common motor signs of PD. For example, akinesia, or an inability to start movement [6–9] is consistent with a problem in movement activation [10].”

We also added another example: “Activation and inhibition are also associated with motor switching and sequencing, which are impaired in PD (Franz, 2006; Cools et al., 1984).” Page 3, Line 59-63.

Page 3, line 64 – Please state the “pre-movement external cueing” which are cited in the references 9-13. Such statement makes it easier for the reader.

Details have been added to this statement on page 3 lines 64-66: “However, deficits in response activation and inhibition associated with PD are complex and not fully understood. For instance, in PD, the latency to initiate movement can be shortened by the use of pre-movement external auditory or visual cueing such as walking with a metronome or use of floor markers (e.g., [12-16]). 

Page 3, line 74 – I think the authors are referring to “Miller”, not colleagues here. If I am correct, please, change this.

This error has been corrected, the line, now on page 3 line 73 now reads: “Franz and Miller [23] found that people with mild to moderate PD demonstrated abnormal force output during movement activation of ‘Go’ responses, as well as problems inhibiting movement of ‘Nogo’ responses in comparison to controls, despite a lack of difference in mean reaction times between PD and control groups.” 

Methods

Subjects – Is this a powered sample, or 36 participants in total were chosen by convenience? Please make this statement.

A power analysis has been added to PAGE 6: line 125 “Power analysis based on Beste et al. (2009; Fig. 2) showed an amplitude difference between PD and control groups of about 8 ±2 µV for both compatible and incompatible control trials for the P300 peak. With an alpha of 0.05 for a single comparison, a sample size of 18 per group yielded power of over 90%. “

Page 6, lines 120-122 – Please, cite the reference about clinical diagnosis of PD. In addition, were these participants diagnosed with idiopathic PD? Please, make it clear.

A reference about clinical diagnosis has been added to page 6; line 122. To clarify that all patients had idiopathic PD the following sentence was added to page 7: Line 144 “All patients had late onset (>55 years at time of diagnosis) idiopathic PD with a history of positive response to dopamine replacement therapy and no alterations in medication for 6 weeks prior to enrollment.”

Page 6, line 133 – How did the authors define “severe tremor”? Did the authors use any tools for that? Please, make it clear. 

The term “severe” was replaced with “persistent,” referring to the fact that it was not well resolved with medication. Persistent tremor was defined as a score > 1 on items 16, 20 or 21 of UPDRS 3 evaluated on medication page 6: line 136). 

Page 10, line 234 – Instead of citing the references 48, 54, please state the details that justify the use of uncorrected multiple comparisons.

We have edited this section (page 11; line 256-261) which now reads: Between-group analysis was corrected with a false discovery rate threshold of p<0.05. For the gamma band, we began with a conservative (multiple comparison corrected) threshold of p<0.05 as a first step, with a more liberal threshold (significant at p<0.05, uncorrected) as a second step if significant effects were not observed at a conservative level. The details of this approach have been described elsewhere [61,67,68]. As we have noted previously [61], the reduced power inherent to higher frequency oscillations often require us to rely on more liberal thresholds with a concomitant increase in risk for type 1 error [61,65]. 

The following statement has also been added to the limitations section (Page 27; line 633-638): “Despite our best efforts to reduce noise, for example enrolling only right-handed subjects with right side disease onset, gamma band activity was noisy, and the more stringent corrected p value yielded minimal consistent activation patterns. Thus our findings on gamma band activity have an increased susceptibility to Type 1 error, though no differences between PD and control participants were observed in the gamma band event related power changes.”

Discussion

I missed 3 important keys in this manuscript: statement of limitations, clinical relevance and conclusion. In regards of limitations, can the authors define any limitation in this study? I understand the importance of these results to understand how PD affects movement control and inhibition, but can the authors work out the clinical relevance of these findings? 

Finally, a conclusion is indeed missing.

As suggested by the reviewer, sections for clinical relevance (Page 26; line 598), limitations (Page 27; line 626) and conclusions (Page 28; line 644) have been added to the manuscript.

Table 1. Is there any statistical between groups for age and gender? Regardless of statistical difference, please state the p-values in the text, or table.

Two sentences were added to the manuscript to provide this information (Page 7; line 142-144): “There were no differences across groups for age F(1, 35) = 0.65, p = 0.43. Gender distribution was not significantly different across groups X2(1, 35) = 2, p = 0.16.” 

Did the authors use UPDRS total score? Please, present the part 3 scores as well. The authors mentioned in the discussion that no correlation was observed between UPDRS and cortical data. Have the authors observed any correlations using the motor score (part 3)? Why did the authors use UPDRS instead of MDS-UPDRS?

Both UPDRS total and part 3 are now included in the table. Correlation? The older version of the UPDRS was used because data collection began before the release of the NDS-UPDRS.

How was the dopamine equivalent calculated? Did authors use Levodopa equivalency daily dosage according to Tomlinson et al., 2010?

To clarify this point, the following statement was added (Page 7; line 151): “Dopamine equivalency for each PD participant’s daily medications was calculated based on conversion factors from Tomlinson et al., 2010.

Reviewer #3: The authors' objectives were to examine PD-associated abnormalities in the magnitude and timing of oscillatory power in cortical networks subserving response activation and inhibition. The paper is interesting and contributes to the understanding of cortical networks subserving response activation and inhibition.

Major reviews:

What do the authors call early Parkinson disease? Table 1 shows that some patients have more than 8 years of illness.

We agree that the word “early” is not accurate and it has been removed from the title and manuscript.

The hypothesis needs further explanation.

To better explain the rationale for our hypothesis, a paragraph on the underlying anatomy has been added to the introduction (Page 4; line 82-90). The introduction now summarizes existing research that motivated the anatomical, latency and frequency band power aspects of our hypothesis: “We tested the hypothesis that oscillatory activity in the frontal cortex subserving these functions had reduced power and delayed onset in PD compared to control participants.” 

How many attempts at practice have been made? Did PD participants and control have the same practice time? Could this have influenced the task's performance during testing?

Training was minimal for both PD and control participants as the task was simple and easy to grasp. To clarify this point, we added details to the description of the training: “Subjects were trained prior to entering the scanner to respond only to the target arrow(s) with their index fingers pressing hand-held button boxes. Stimuli consisted of a series of stimuli that contained 2 trials of each trial type. All subjects executed this trial run once before scanning following which they reported understanding the task.” Page 7; line 159- page 8; line 165

Were patients evaluated ON or OFF? What does this interfere with the results?

To clarify that all study activities were preformed on medication, the statement about evaluation was changed to the following: “PD participants took their prescribed medication in the morning prior to study participation and performed the experiment in their best ON medication state.” (Page 7; line 146-148)

In addition, the following statement was added to indicate that the UPDRS was also performed on medication: “Participants with PD were given the Unified Parkinson Disease Rating Scale (UPDRS) on medication on a separate day from brain imaging.” (Page 6; line 138-139)

A statement about the impact of ON medication evaluation was added page 26; lines 601-603. 

Were the conditions randomized or presented in blocks? Was the task learned during the experiment?

The statement, “Trial types were presented in random order” was added to end of the section describing the trials (Page 9; lines 195-196). Regarding learning, the task was very simple, and we did not observe an error pattern (fewer errors over time) that was indicative of learning. 

Discuss the implications of having found differences in cortical activity despite the same task performance.

The discussion of the interpretation of differences in cortical activity with similar task performance have been expanded. The paragraph on the distribution of abnormal network activity in response to movement activation now includes a discussion of a pathological, compensatory and response strategy interpretation (Page 18; line 446). The possibility of compensation is also addressed in the discussion of frequency abnormalities (Page 25; line 565-579) and in the conclusions (page 28; line 645).

A Conclusion section would be important.

We agree, and a conclusions section has been added (Page 28; line 644).

Minor review:

- Place the units in the tables

Units have been placed in the tables

- what is the patients' H&Y?

Median H&Y has been added to Table 1

- are the values in table 3 reported as mean and standard deviation? 

Yes, this information has been added to the legend.

- check the link to the page (http://www.sourcesignal.com/dataeditor.html) that is not working

The link has been updated. The correct link is (https://www.ctf.com/products).

- What was considered a wrong attempt?

An error was defined as a unilateral button press of the incorrect button, a unilateral response when a bilateral response was required or a bilateral response when a unilateral response was required. This definition was added to page 9, line 206.

---

## [Decision Letter · Decision Letter 1]

3 Mar 2021

PONE-D-20-21985R1

Cortical Oscillatory Dysfunction in Parkinson Disease During Movement Activation and Inhibition

PLOS ONE

Dear Dr. Disbrow,

Thank you for submitting your manuscript to PLOS ONE. After careful consideration, we feel that it has merit but does not fully meet PLOS ONE’s publication criteria as it currently stands. Therefore, we invite you to submit a revised version of the manuscript that addresses the points raised during the review process.

Dear authors,

One reviewer indicated some aspects that you might improve the manuscript. I call attention to the discussion. I agree totally with the reviewer, and the discussion should be more straight writing, reducing its size. Please address all suggestions in the manuscript.

We look forward to receiving your revised manuscript.

Kind regards,

Fabio A. Barbieri, PhD

Academic Editor

PLOS ONE

Journal Requirements:

Reviewers' comments:

Reviewer's Responses to Questions

**Comments to the Author**

1. If the authors have adequately addressed your comments raised in a previous round of review and you feel that this manuscript is now acceptable for publication, you may indicate that here to bypass the “Comments to the Author” section, enter your conflict of interest statement in the “Confidential to Editor” section, and submit your "Accept" recommendation.

Reviewer #2: All comments have been addressed

Reviewer #3: All comments have been addressed

2. Is the manuscript technically sound, and do the data support the conclusions?

Reviewer #2: Partly

Reviewer #3: Yes

3. Has the statistical analysis been performed appropriately and rigorously? 

Reviewer #2: Yes

Reviewer #3: Yes

4. Have the authors made all data underlying the findings in their manuscript fully available?

Reviewer #2: Yes

Reviewer #3: Yes

5. Is the manuscript presented in an intelligible fashion and written in standard English?

Reviewer #2: Yes

Reviewer #3: Yes

6. Review Comments to the Author

Reviewer #2: The authors improved the manuscript and addressed the concerns I raised. Although, there are some parts of the manuscript which the authors should carefully address.

Page 6: was there any inclusion and exclusion criteria for healthy controls? Please, state them.

Page 6, end of last paragraph: “Using an alpha of 0.05 for a single comparison, a sample size of 18 per group yielded power of over 90%”

What statistical test the authors refer to?

Page 12 end of last paragraph: “analysis of variance (ANOVA) with t-test post hoc analysis”.

The authors should correct this to “… (ANOVA) with post hoc analysis using a p value of 0.05”

Caption for Table 3. Please, replace “#” with the word “number” or “amount”.

The authors first introduced the ROI in the results section. I suggest the authors introduce the regions of interest in the methods section.

Why do the authors present p-values for only some of the differences, and not for all? Please, present p-values for all comparisons.

Discussion – The discussion is too large (8 pages) and somewhat repeat the results. The authors have been comparing their results with literature rather than deeply discussing the meaning of the results found in the study. Please, update the discussion making it more concise and easier to follow.

Please see the manuscript from Pelicioni and colleagues (2020) where they discuss how people with PD exhibit reduced cognitive and motor cortical activity when undertaking complex stepping tasks requiring inhibitory control. Even though the authors have written the clinical implication in their most updated version of the manuscript, I still think the paper has many technical aspects and not enough clinical implications, and how these results can be translated to the understanding of motor impairment (such as inhibition impairments) in people with PD.

Reference:

People With Parkinson’s Disease Exhibit Reduced Cognitive and Motor Cortical Activity When Undertaking Complex Stepping Tasks Requiring Inhibitory Control. PHS Pelicioni, SR Lord, Y Okubo, DL Sturnieks, JC Menant. Neurorehabilitation and Neural Repair 34 (12), 1088-1098

Reviewer #3: This resubmission is improved with the addition of necessary details in the Methods and a more fulsome Discussion.

7. PLOS authors have the option to publish the peer review history of their article (what does this mean?). If published, this will include your full peer review and any attached files.

Reviewer #2: No

Reviewer #3: **Yes: **Daniel Boari Coelho

---

## [Author Response · Author response to Decision Letter 1]

21 Jul 2021

We would like to thank the reviewers for their thoughtful comments on this manuscript. All suggestions have been incorporated into the latest version. Most notably, the discussion has been extensively edited based on reviewer comments to be more accessible. Specific responses are described below.

Page 6: was there any inclusion and exclusion criteria for healthy controls? Please, state them.

The inclusion and exclusion criteria for controls have been added on page 6, line 133. 

Page 6, end of last paragraph: “Using an alpha of 0.05 for a single comparison, a sample size of 18 per group yielded power of over 90%” What statistical test the authors refer to? 

The line now reads: Using an alpha of 0.05 for a single comparison, a sample size of 18 per group yielded power of over 90% for an independent, 2 sample t-test comparing PD and control groups. Similar results were obtained for a power analysis for latency measures. 

Page 12 end of last paragraph: “analysis of variance (ANOVA) with t-test post hoc analysis”. The authors should correct this to “… (ANOVA) with post hoc analysis using a p value of 0.05”

The text has been changed as indicated.

Caption for Table 3. Please, replace “#” with the word “number” or “amount”.

The text has been changed as indicated.

The authors first introduced the ROI in the results section. I suggest the authors introduce the regions of interest in the methods section.

We have added the following sentence to the last paragraph of the introduction: “to measure anatomical, at the level of the cortical field region of interest, and physiological, specifically electrophysiological data.” A sentence about the ROI definition procedure and a reference have also been added to the methods on page 11, line 261. “Regions of interest (ROI) based on Brodmann nomenclature were derived from MNI coordinates in the normalized brain [68].”

Why do the authors present p-values for only some of the differences, and not for all? Please, present p-values for all comparisons. 

The p values for the ANOVA’s were provided in the results while the p value for the NUTMEG analysis were provided in the methods, which, as indicated by the reviewer, was odd. The NUTMEG p values are now included in the results as well to clarify the statistics.

Discussion – The discussion is too large (8 pages) and somewhat repeat the results. The authors have been comparing their results with literature rather than deeply discussing the meaning of the results found in the study. Please, update the discussion making it more concise and easier to follow.

This comment was very helpful and the discussion has been extensively edited. The discussion has been reduced from just over 10 pages to 8 pages including additions suggested by reviewers. The discussion of previous work was reduced and put in context, making the discussion more concise and easier to follow. We also created a section on compensation (page 23, line 533) to improve the organization and flow of the discussion.

Please see the manuscript from Pelicioni and colleagues (2020) where they discuss how people with PD exhibit reduced cognitive and motor cortical activity when undertaking complex stepping tasks requiring inhibitory control. Even though the authors have written the clinical implication in their most updated version of the manuscript, I still think the paper has many technical aspects and not enough clinical implications, and how these results can be translated to the understanding of motor impairment (such as inhibition impairments) in people with PD.

We read the recommended reference with interest. The clinical significance section has been amended along the lines of this paper, including a reference to our work on neurorehabilitation of movement activation, and a modified statement about the clinical implications of inhibition deficits in PD. The end of the clinical implications section now reads: “Data on abnormal cortical activity may inform intervention strategies [156–158]. For example, cognitive neurorehabilitation targeting movement activation has shown some success [157,158]. Future studies of neurorehabilitation using MEG could yield clinically significant insight into the plasticity subserving recovery of function. Impairments of movement inhibition, though clearly present in PD, were described only recently [2] and, arguably, make subtle and complex contributions to classic motor signs such as gait or intricate stepping abnormalities [156]. As in most complicated diseases, the picture of Parkinson's disease becomes clearer with accumulation of evidence at all levels of the neural-cognitive system.”

---

## [Editor Report · Decision Letter 2]

9 Sep 2021

Cortical Oscillatory Dysfunction in Parkinson Disease During Movement Activation and Inhibition

PONE-D-20-21985R2

Dear Dr. Disbrow,

We’re pleased to inform you that your manuscript has been judged scientifically suitable for publication and will be formally accepted for publication once it meets all outstanding technical requirements.

Kind regards,

Fabio Augusto Barbieri, PhD

Academic Editor

PLOS ONE
---

## [Editor Report · Acceptance letter]

15 Feb 2022

PONE-D-20-21985R2 

Cortical Oscillatory Dysfunction in Parkinson Disease During Movement Activation and Inhibition 

Dear Dr. Disbrow:

I'm pleased to inform you that your manuscript has been deemed suitable for publication in PLOS ONE. Congratulations! Your manuscript is now with our production department. 

Kind regards, 

on behalf of

Fabio Augusto Barbieri 

Academic Editor

PLOS ONE